# Extracellular Vesicles as Novel Drug-Delivery Systems through Intracellular Communications

**DOI:** 10.3390/membranes12060550

**Published:** 2022-05-25

**Authors:** Yasunari Matsuzaka, Ryu Yashiro

**Affiliations:** 1Division of Molecular and Medical Genetics, Center for Gene and Cell Therapy, The Institute of Medical Science, University of Tokyo, Minato-ku 108-8639, Tokyo, Japan; 2Administrative Section of Radiation Protection, National Institute of Neuroscience, National Center of Neurology and Psychiatry, Kodaira 187-0031, Tokyo, Japan; ryuy@ncnp.go.jp; 3Department of Infectious Diseases, Kyorin University School of Medicine, 6-20-2 Shinkawa, Mitaka-shi 181-0004, Tokyo, Japan

**Keywords:** adeno-associated virus, extracellular vesicles, intracellular communication, membrane vesicle production, non-coding RNAs

## Abstract

Since it has been reported that extracellular vesicles (EVs) carry cargo using cell-to-cell comminication according to various in vivo situations, they are exprected to be applied as new drug-delivery systems (DDSs). In addition, non-coding RNAs, such as microRNAs (miRNAs), have attracted much attention as potential biomarkers in the encapsulated extracellular-vesicle (EV) form. EVs are bilayer-based lipids with heterogeneous populations of varying sizes and compositions. The EV-mediated transport of contents, which includes proteins, lipids, and nucleic acids, has attracted attention as a DDS through intracellular communication. Many reports have been made on the development of methods for introducing molecules into EVs and efficient methods for introducing them into target vesicles. In this review, we outline the possible molecular mechanisms by which miRNAs in exosomes participate in the post-transcriptional regulation of signaling pathways via cell–cell communication as novel DDSs, especially small EVs.

## 1. Introduction

Recently, because the safety and efficacy of the drug and its proper transmission to the target organ are important factors, the development of various drug-delivery systems (DDSs) has been energetically promoted (Table 1). Expections are especially growing for new DDSs using extracellular vesicles (EVs) via cell-to-cell communications. In addition, emerging evidence from the modulatory role of non-coding-RNA (ncRNA) fetuses has attracted attention to their potential as diagnostic biomarkers for liquid biopsies and therapeutic applications [1,2,3]. Based on their length, ncRNAs can be subdivided into two main categories: short ncRNAs (<200 nucleotides) and long ncRNAs (>200 nucleotides) [4,5,6]. Within short ncRNAs, circulating microRNAs, which are single-stranded RNAs that are approximately 22 nucleotides long, can be used as potential biomarkers for diagnosis, prognosis, and therapeutics in various diseases, but are also delivered for intracellular communications through EVs-encapsulated forms [7,8].

EVs are a heterogeneous group of membrane-enclosed structures of varying sizes and different origins that are released from almost every cell type [9,10]. EVs, classified by size, are categorized as such: exosomes (30–150 nm diameter, 100,000× *g* sedimentation speed), which are released from the fusion of multivesicular bodies (MVBs) with the plasma membrane; large EVs, such as microvesicles, also called outer membrane vesicles, ectosomes, or shedding vesicles (100 to 1000 nm diameter, 10,000× *g* sedimentation speed), which are derived from the outward budding of the cell membrane when the curtain is torn off; apoptotic bodies, which fall off during the process of cell apoptosis (50 to 5000 nm diameter, 2000× *g* sedimentation speed) [11,12,13,14,15,16,17]. EVs are also categorized based on their content, biogenesis, functions, sedimentation speed during stepwise centrifugation of the sample, as well as by their secretory origins, such as the plasma membrane and endosomes (Figure 1) [18,19,20].

Furthermore, it has recently become clear that there are subtypes with different contents, even specific EV subtypes [21,22]. However, it remains unclear how the heterogeneity of EVs affects their functions. Additionally, because the markers that clearly distinguish exosomes and microvesicles are not specified, they have recently been fractionated into large, intermediate-sized, and small EVs according to the sedimentation rate [23,24,25]. In fact, recently, a novel type of small nonmembranous nanoparticle (<35 nm), called an exomere, was identified [23,24,25]. Small EVs exhibit a cup-shaped morphology, whereas the exomeres show a dot-shaped morphology, ranging in size from 39 to 71 nm, which were found to be enriched with Argonaute proteins (Ago1, Ago2, and Ago3), esterified cholesterol (4:1 esterified/unesterified), and more small RNAs compared with the small EVs, but an approximately four-fold reduction in total lipid content relative to small EVs was also found [23,24,25]. Furthermore, a new population of nanoparticles, referred to as supermeres (supernatant of exomeres), which are morphologically and structurally distinct from exomeres and exhibit smaller heights and diameters than exomeres, have recently been reported [26,27]. The supermere, which is highly enriched with glycolytic enzymes, transforming growth factor beta-induced (TGFBI), miR-1246, cellular–mesenchymal–epithelial transition factor (MET), and Ago2, showed different compositions of proteins and ribonucleic acid (RNA) compared to EVs and exosomes. Surprisingly, the majority of extracellular RNA is associated with supermeres rather than with small EVs and exomeres. Additionally, the supermere exhibits markedly greater uptake in vivo than small EVs and exomeres. Furthermore, supermeres and exomeres display significantly slower cellular uptake than small EVs. 

First, the Greek word “exosomes” comes from “Exo”(“outside”) and “soma”(“body”) [28]. Recently, while investigating the maturation of sheep reticulocytes into erythrocytes by horizontally transferring proteins, lipids and nucleic acids, exosomes, which were assumed to be cellular debris or garbage released from reticulocytes and considered as signs of cell death, were observed to influence crucial regulatory roles as intracellular communicators as well as in accelerating interorgan crosstalk in various pathophysiological processes, including cancer, immune responses, inflammation, infection, neurodegenerative diseases, metabolic diseases, and cardiovascular diseases, in both parent and recipient cells [29,30,31,32,33]. These EV-encapsulated contents could be potential biomarkers and therapeutic targets as alternatives to liposome-mediated drug-delivery systems for diverse diseases.

## 2. Biogenesis and Release of Exosomes

### 2.1. Endosomal Sorting Complexes Required for Transport (ESCRT)-Dependent and -Independent Pathways

Biogenesis of exosomes begins with the inward budding of the plasma membrane by endocytosis of the plasma membrane to form multivesicular endosomes (MVEs), which mature into MVBs that incorporate different cytosolic components such as proteins, nucleic acids, and lipids by the intraluminal vesicles (ILVs) through the limiting membrane of late endosomes, where the invagination process of the ILVs determines the specific compositions of proteins and lipids on the vesicle membrane, leading to distinctive molecular profiles [34,35,36]. Furthermore, ESCRT-dependent and ESCRT-independent (neutral sphingomyelinase 2/Sphingomyelin phosphodiesterase 3, nSMase2/Smpd3) pathways have been identified as the pathway for protein sorting into exosomes and exosome biogenesis via hydrolysis of sphingolipids to ceramides on the cell membrane [37,38]. The ESCRT family can regulate cargo loading into exosomes through post-translational modifications of protein ubiquitination [39]. It has been reported that suppression of the ESCRT-dependent pathway inhibits the release of small-exosome-encapsulated ADAM metallopeptidase domain 10 (Adam10) and *MET* Proto-Oncogene, Receptor Tyrosine Kinase (Met), while secretion is not affected by nSMase2/Smpd3 [40]. This suggests that different small exosomes with different compositions of protein cargos may be produced by the ESCRT complex and the nSMase2/Smpd3 protein owing to the specificity and affinity of these membrane proteins to specific membrane domains. In addition, the ESCRT-dependent pathway works in concert with heparan sulfate proteoglycan, syndecan, and its adapter protein, syntenin-1, and is responsible for the formation of syntenin-1-containing exosomes [41,42,43,44,45]. Syntenin directly interacts with ALG-2-interacting protein X (ALIX), an auxiliary component of the ESCRT machinery [41,45,46]. The syntenin–ALIX complex links syndecans and syndecan cargo to the ESCRT budding machinery at the MVBs. Interestingly, this pathway is activated by heparanase, which degrades heparan sulfate, indicating that glycan modifications modulate exosome biogenesis [44]. Some tetraspanins in the biogenesis of exosomes determine the abundance of ESCRT components and associated proteins, such as tumor susceptibility gene 101 (TGS101) and ALIX on the exosome membrane [47]. The MVEs fuse with lysosomes to degrade their cargo, including their ILVs, by hydrolases [48,49,50]. However, after the fusion of late endosomes and MVBs with cell membranes, their ILVs are released into the extracellular space as exosomes, which are detected in different biological fluids, such as cerebrospinal fluid, saliva, blood, breast milk, amniotic fluid, seminal fluid, and urea (Figure 2) [51,52,53,54,55,56,57]. 

### 2.2. Proteins, Peptides, Lipid, and Nucleic Acids in Exosomes

In addition, the composition of proteins, peptides, and nucleic acids in exosomes is independent of donor cell type [58]. Lipid compositions in exosomes, which are commonly enriched with cholesterol and sphingomyelin, and minimal amounts of lecithin and phosphatidylethanolamine, primarily depend on exosome-producing cells [58,59,60,61,62]. Furthermore, it has been revealed that lean mice become insulin resistant due to the administration of exosomes isolated from the feces of obese mice that were fed a high-fat diet (HFD) and of type II diabetic patients [63,64,65]. An HFD alters the lipid composition of exosomes [63]. In addition, phosphatidylethanolamine is abundant in the exosomes of lean animals [63]. In contrast, phosphatidylcholine (PC) is abundant in the exosomes of obese animals [63]. Mechanistically, it was suggested that exosomes in the intestine of obese animals are taken up by macrophages and hepatocytes, leading to the inhibition of the insulin-signaling pathway. Exosome-derived PCs bind to and activate aromatic-hydrocarbon receptors (AhRs) and suppress the expression of genes essential for the activation of insulin-signaling pathways, such as insulin receptor substrate 2 (IRS-2) and its downstream genes, phosphatidylinositol-3 kinase (PI3K) and Protein Kinase B/thymoma viral proto-oncogene 1 (PKB/AKT) [63]. These reports reveal that exosomes produced by an HFD may contribute to the development of insulin resistance. 

Transmission electron microscopy (TEM) with negative staining and cryo-TEM indicated that exosomes have cup-shaped and perfectly spherical structures, respectively [66]. The selective enrichment of miRNAs, which is abundant among small RNA families that are encapsulated in exosomes, depends on their size, origin, and transcription by RNA polymerase III [67,68,69,70]. miRNAs are transcribed by RNA polymerase (Pol) II into primary miRNAs, which Drosha/DGCR8 processes into precursor miRNAs [71,72,73,74]. The pre-miRNA is transported from the nucleus into the cytoplasm by exportin 5 and then processed by Dicer/TRBP into an miRNA duplex [75,76,77]. After the formation of a duplex complex with the RNA-induced silencing complex (RISC), mature microRNAs are packed into exosomes in different manners: I) the miRISC-related pathway, II) miRNA motif-sumoylated heterogeneous nuclear ribonucleoproteins (hnRNPs)- and the synaptotagmin-binding cytoplasmic-RNA-interacting-protein (SYNCRIP)-dependent pathway, which recognizes the short sequence motifs GGAG and GCUG in the 3′-miRNA sequence, and III) the ceramide-dependent pathway [78,79,80,81,82,83,84]. In addition, RNA-binding protein (heterogeneous nuclear ribonucleoprotein A2B1: hnRNPA2B1) and Y-box protein are involved in the sorting of miRNAs in the EVs [85,86].

## 3. EV-Mediated DDS

### 3.1. microRNAs in EV-Mediated DDS

The expansion of drug-discovery modalities has been remarkable in recent years. In particular, nucleic-acid drugs can directly act on gene regulation [87,88,89]. Compared with conventional small-molecule drugs, biopharmacology makes it easier to design drugs that specifically act on various molecules. For example, short double-stranded RNA (siRNAs) can degrade RNA with complementary sequences [90,91]. Furthermore, the design of a double-stranded RNA with a base sequence corresponding to the target gene makes it possible to specifically suppress the expression in the target gene [92]. Since there is no difference in the basic physiochemical properties of different siRNAs in terms of biopharmacology, it is unlikely that some problems related to physicochemical properties, such as solubility, stability, and pharmacokinetics—including absorption and metabolism—caused by differences in physical properties between compounds will occur. Therefore, they have attracted attention as a next-generation modality that sets them apart from other pharmaceutical products. However, nucleic-acid drugs have problems such as low intracellular-delivery efficiency due to low cell-membrane permeability, their large molecular weight, and elimination by the cellular immune system [93,94,95,96,97]. To solve these problems, the development of a DDS using the chemical modification of nucleic acids and liquid nanoparticles is underway. [98,99] In addition, liposome-based drugs have been approved, and research and development of novel DDS strategies for nucleic-acid drugs is accelerating [100,101,102]. However, because liposomes are non-natural substances in the human body, problems such as immunogenicity and biotoxicity remain. On the other hand, EVs are natural substances with low immunogenicity and biotoxicity [103,104,105,106,107,108,109,110,111,112,113,114,115,116]. Cancer-cell-derived EVs include cancer antigens according to individual cancer types, and the anti-tumor immune response is induced by the uptake of dendritic cells, which are the main antigen-presenting cells [117,118,119,120]. In particular, they have the potential to be used as a new immunotherapy for cancers for which cancer antigens have not yet been identified. However, even if cancer-cell-derived EVs are used alone, activation of DCs, which are essential for inducing an anti-tumor immune response, cannot be expected. 

### 3.2. Modification of EVs as DDS Methods

When trying to develop DDSs with EVs, which are natural drug carriers, there are two methods: loading the drug directly into the recovered EVs [121,122,123], and loading drugs into EVs by modifying the function of secretory cells [124,125,126]. In particular, the method of imparting functions to EVs that are extracted by manipulating the cells described above is significantly different from the DDS technology that is applied to conventional artificial nanoparticle preparations. It was also revealed that CpG DNA-modified EVs showed a strong anti-tumor immune response by reacting with biotin derivatives of CpG DNA, which is an immunostimulatory nucleic acid, with the EVs modified by the fusion of protein with streptavidin showing binding to biotin and lactadherin, which is an EV-migrating protein [117,127].

In addition, the secretion of EVs is not limited to mammals, and has already been found in plants and microorganisms [128,129,130]. It is thought that there is a mechanism by which molecules with immunomodulatory ability are secreted from microorganisms that have beneficial effects to the outside of the cell and delivered to the target cell of the living body [131]. When various EVs derived from bifizus, butyric-acid, and lactic-acid bacteria were added to two types of immune cells, mouse macrophage-like-cell line RAW264.7 and dendritic-cell line DC2.4, the production of inflammatory cytokines, tumor necrosis factor-α (TNF-α) and interleukin 6 (IL-6) was increased [132,133,134,135]. This report confirmed that EVs derived from probiotics have immunostimulatory abilities. 

### 3.3. Recovery of EVs as DDSs

In order to use EVs as DDSs, recovery-system operations are required because EVs are contained in the supernatant of the culture medium in cultured cells and serum in various concentrations and forms [136,137]. Although several EV-recovery methods have been reported, they have both advantages and disadvantages in terms of yield and use [20,138]. (1) The ultracentrifugation method removes impurities from cell-culture supernatants and blood samples by stepwise centrifugation [139,140]. Generally, debris such as cells are removed by centrifuging the culture supernatant and serum at 10,000× *g* for 30 min. They are then centrifuged at 100,000× *g* for 70 min to precipitate EVs and proteins. PBS is then added to the pellet, which is washed and centrifuged at 100,000× *g* for 70 min to precipitate the EVs. The ultracentrifugation method is easy to operate and suitable for large-scale recovery, but it is inferior in terms of the recovery rate. (2) The polyethylene-glycol (PEG) precipitation method uses PEG as a precipitating agent [141,142]. The PEG reagent with a molecular weight of 6000 or 8000 is mixed with a biological sample to a final concentration of 10% (*w/v*), allowed to stand at 4 °C, and then centrifuged at 1500 to 3000× *g* for 300 to 60 min. The resulting precipitate contains the EVs. The advantages of this method are that it does not require an ultracentrifuge or expensive and special reagents, and its operation is as simple as the ultracentrifugation method. However, the particle-size distribution of the obtained EVs is wide, and it cannot be ruled out that particles larger than EVs may be included. (3) The immunoprecipitation method is a recovery method that uses an antibody that recognizes EV membrane proteins [141,143]. Magnetic beads, to which an antibody is bound, can be added to a sample, and exosomes can be easily separated and recovered using a magnet. Because the IP method is based on the antigen–antibody reaction, it is possible to recover high-purity and specific EVs. This method can be used for unknown samples; however, it is difficult to recover EVs for which the target antigens have not been found. (4) Density-gradient centrifugation is a method of separating relatively low-density EVs from other vesicles and particles by combining a sucrose density gradient and a sucrose cushion [144]. Although this separation method can isolate high-purity EVs, it is time-consuming and requires sophisticated techniques. 

### 3.4. Preparation of EVs as DDSs

Furthermore, drug delivery using the substance-transport capacity of EVs can be broadly divided into two types. One is the pre-loading method, which modifies EV-producing cells, and the other is the post-loading method, in which the recovered EVs are chemically or physically loaded with a drug [145,146,147,148,149]. In the post-loading method, at first, the electroporation method is used to introduce a substance using an electric current [126]. The electric field causes partial membrane disruption and improves membrane permeability, allowing substances to enter the EVs. While this method can be applied to any EV, there is a concern that RNA and EVs will aggregate due to electric-field treatment. Second, sonication is a method of introducing a drug into the membrane by temporarily impairing its integrity by ultrasonic treatment [126]. However, this method also detects non-spherical EVs owing to changes in the membrane structure. Third, saponin, a surfactant molecule, can form a complex with cholesterol in the lipid membrane to form pores and improve the permeability of hydrophilic molecules [150]. However, saponins have been reported to have hemolytic activity in vivo and should be carefully considered before use. 

In the above-mentioned method of loading a drug into EVs, due to the necessity of isolating and purifying the EVs using both methods, it is difficult to avoid a significant decrease in the recovery rate and degeneration of the EV membrane during the process [151,152]. Yamayoshi et al. reported a new DDS strategy for intracellularly delivering drugs using EVs without isolation and purification [153]. This new method indicates that a complex of an antibody-bound nucleic acid consisting of a monoclonal antibody targeting CD63, which is an EV membrane surface protein, and anti-miR is taken up into cells by being associated with EVs and inhibits the function of miRNA in the cytoplasm. Furthermore, in cancer-bearing mice that were subcutaneously transplanted with cells treated with this antibody–anti-miR complex, the tumor volume was significantly smaller than that in the non-treated group, suggesting that a tumor-forming-inhibition ability was observed. In addition, it has been observed that tumor volume is reduced when this complex is administered to cancer-bearing mice via the tail vein. This report demonstrated the effectiveness of functional molecules targeting EV–miRNAs, which exert advanced functions in trace amounts and play a major role in bioregulatory systems. Moreover, this method can introduce not only anti-miRs but also antisense nucleic acids, siRNAs, aptamer-targeting mRNA, small molecule compounds, and nucleic acids. Therefore, the development of new molecular-targeted drugs targeting not only EV–miRNA but also various molecules contained in EVs is expected. 

Furthermore, proteinase K was observed to reduce uptake efficiency by removing proteins from the surface of exosomes or uptake cells [154,155]. The recognition of molecules on the surface of exosomes by uptake cells is important for the mechanism, in which the surface molecule of an exosome depends on the type of producing cell, and the type of molecule on the uptake-cell side also depends on the type of uptake cell. It has been shown that the combination of producing cells and uptake cells is an important factor that determines the cellular-uptake characteristics of exosomes [156,157]. Further, it has been shown that uptake occurs by fusion of the lipid membrane between the membrane of exosomes and the cell membrane of uptake cells [156]. Moreover, the lipid composition is important because the addition of filipin reduces the efficiency of membrane fusion by changing the lipid composition of the uptake-cell membrane [156]. In addition, it has also been reported that uptake changes depending on the state of exosome-producing or uptake cells and the surrounding environment, in addition to the types of exosome-producing cells and uptake cells [34]. Exosomes recovered under acidic conditions change their lipid composition and are more easily taken up by cells [158]. It has been reported that exosome uptake efficiency increases under acidic conditions. Moreover, since treatment with TNF-α increases the expression level of the integrin-recognition molecule ICAM on the fibroblast surface of uptake cells, the uptake of B-cell-derived exosomes by fibroblasts is increased due to the expression of integrin β1 and β2 on the surface of the exosome [159]. Regarding the uptake mechanism of exosomes by human plasmacytoid dendritic cells (pDC) and conventional dendritic cells (cDC), it became clear that the uptake rate in cDCs was faster than that in pDC. The uptake by cDC has been shown to be due to endocytosis, whereas the small contribution of endocytosis to the uptake of exosomes by pDC indicates that this uptake may be slow [160]. It has also been shown that phagocytic cells such as mouse macrophage-like-cell line RAW264.7 cells and *human-leukemia-T-cell* line Jurkat cells have high exosome uptake efficiency, but non-phagocytic cells such as mouse fibroblast-like-cell line NIH3T3 cells and fibroblast-like-cell lines derived from monkey kidney-tissue COS-7 cells take up few exosomes, and this uptake is reduced by inhibitors of actin polymerization and PI3K [161]. Endocytosis is the main mechanism of exosome uptake, and inhibitors of clathrin-dependent endocytosis and macropinocytosis reduce uptake [162,163]. 

Additionally, macrophages phagocytose apoptotic cells through recognition of the externalization of the phosphatidylserine (PS)-induced cleavage of intracellular proteins by activated caspase as “eat me signals”. The PS on the surface of the exosome may modulate the “eat me signal” in the apoptotic pathway through molecular regulation via the cargo (Figure 3) [164,165]. 

## 4. EV-Encapsulated Adeno-Associated Virus

Adeno-associated virus (AAV) is a virus with a diameter of approximately 20–26 nm, which is extremely physicochemically stable with no envelope and is classified in the genus *Dependoparvovirus* of the family *Parvoviridae* [166,167]. More than 100 types of AAV are known, but the most advanced AAV is the type 2 AAV (AAV2) [168]. The AAV2 capsid structure is composed of three types of proteins, VP1 (82 kDa), VP2 (65 kDa), and VP3 (60 kDa), assembled at a ratio of 1:1:10 (Figure 4) [169]. The genome is a single-stranded DNA consisting of approximately 4.7 nucleotides, with plus and minus strands mixed in approximately the same population, where the 145 nucleotides at both ends of the genome form a T-shaped hairpin structure referred to as inverted terminal repeats (ITRs), which becomes the starting point of duplication [170]. The AAV genome contains two genes, *rep* and *cap*, which encode nonstructural proteins and capsid proteins, respectively. AAV can only grow in the presence of helper viruses such as adenovirus and herpesvirus. When a cell is infected with AAV alone, its genome is specifically integrated into the adeno-associated-virus-integration-site-1 (AAVS1) region of chromosome 19, 19q13.42, resulting in latent infection [171]. However, it is extremely unlikely that the transgene loaded on the viral genome of the recombinant AAV, referred to as the AAV vector, will be integrated into the chromosome by the AAV vector, and it is believed that the transgene is basically extrachromosomal. The transgene is a single-stranded DNA in the AAV vector, but it is converted to double-stranded DNA in the cell, where the transgene is expressed. The expression of genes introduced into neurons by the AAV vector has been shown to persist for long periods of time [172]. Furthermore, no specific disease associated with AAV2 infection has been reported, and its non-pathogenicity has attracted attention as a useful and safe gene-therapy vector for various diseases. Moreover, AAV has a wide host range within humans and primates, and infects various proliferating and non-proliferating cells [171]. It has been reported that heparan sulfate, *fibroblast-growth-factor* (FGF) receptor, and αVβ5 integrin are known as receptors or co-receptors for AAV2 [173,174,175]. In addition, AAV4 and AAV5 have been shown to bind to sialic acid, and the platelet-derived-growth-factor (PDGF) receptor acts as a receptor for AAV5 [176].

On the other hand, the safety and efficacy of AAV vectors need to be improved due to the occurrence of deaths in clinical trials caused by single high-dose administration, imperfect organ orientation, or the presence of neutralizing antibodies of AAV vectors that bind to the AAV capsid and inhibit the interaction with target cells, resulting in a reduction of AAV-transduction efficacy, which is a major clinical barrier. Previously, it was reported that AAV is naturally secreted via the interior or surface of EVs into the extracellular space in two types of forms, EV-encapsulated or EV-associated AAVs (EV–AAVs), which can be taken up with high transduction efficacy by target cells [177,178,179,180,181,182,183,184]. Furthermore, the EV–AAV vector can be protected from neutralizing antibodies in vivo and in vitro compared to non-encapsulated or associated AAVs [185,186]. Taken together, the EV–AAV will be a possible new therapeutic strategy in various diseases.

## 5. Conclusions

EVs are membrane-enclosed structures with heterogeneous sizes and origins. EVs have attracted much attention as novel potent DDSs owing to the incorporation of molecular information such as proteins, lipids, and nucleic acids via intracellular communications. The diversity of EV functions is dependent on their composition, such as the type and amount of proteins and lipids on the membrane or nucleic acids, including miRNAs, within the EVs. Therefore, the isolation and purification standards of EVs are important for the establishment of therapeutic methods using the EV-mediated modification of target genes. Furthermore, a virus-vector-encapsulated EV showed very high transduction efficacy and protection from neutralizing antibodies compared to free virus vectors. Thus, EV-mediated transduction of the contents, including virus vectors or miRNAs, will be a powerful tool for various disease treatments as a novel DDS.

## Figures and Tables

**Figure 1 membranes-12-00550-f001:**
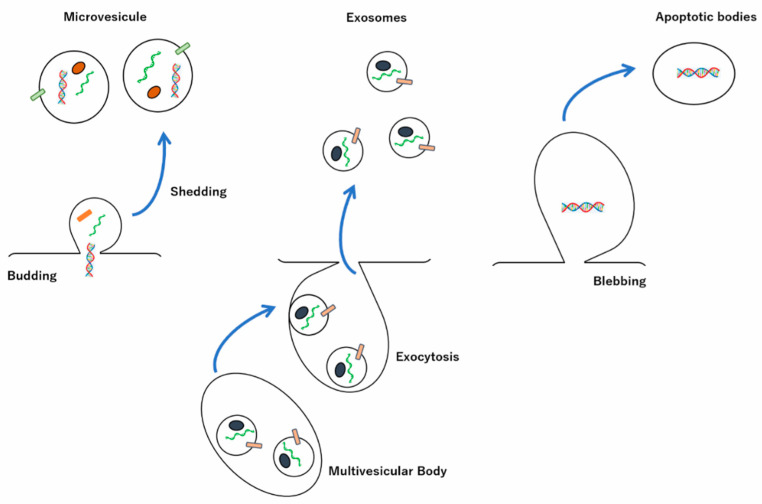
Secretions of extracellular vesicles (EVs). EVs are divided into three main populations according to their diameters: exosomes at 30 to 150 nm, microvesicles (MVs) at 100 to 1000 nm, and apoptotic bodies at 50 to 5000 nm. The exosome biogenesis involves the formation of multivesicular bodies and transfer to the plasma membrane, where the exosomal contents, proteins, and miRNAs are released via their fusion. The formation of both MVs and apoptotic bodies is accompanied by budding and blebbing of the cell membrane to pinch off new vesicles, respectively.

**Figure 2 membranes-12-00550-f002:**
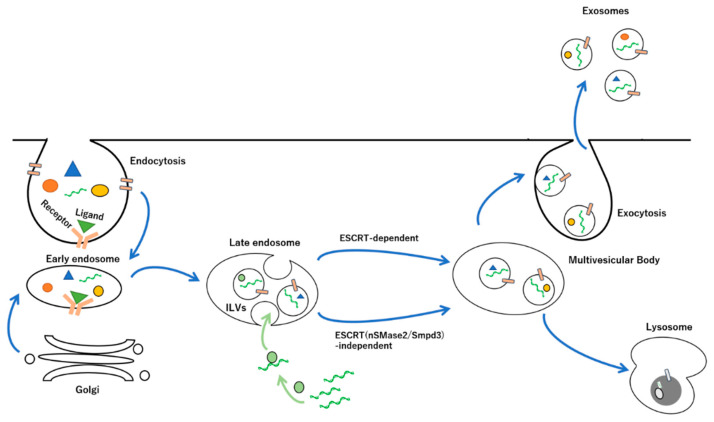
Mechanisms of exosome biogenesis and secretion. Upon the binding of ligand with exosome, the endocytosis process involves inward budding of the cell membrane, which occurs at lipid rafts containing common membrane proteins, such as tetraspanins (e.g., CD9, CD81, CD63, etc.), MHC class I and class II, and adhesion molecules (e.g., integrins, cadherins, etc.). The internalized cargoes, including proteins, nucleic acids, and lipids are sorted into early endosomes, which mature into late endosomes. The intraluminal vesicles (ILVs) of the late endosomes are formed through budding from the perimeter membrane into the endosome lumen following the encompassing the bioactive molecules by ESCRT-dependent or -independent pathways. Multivesicular bodies (MVBs) containing some ILVs are selectively led into two different pathways, i.e., lysosomal degradation or secretion of exosomes toward the extracellular space through the fusion of lysosomes, or the plasma membrane in the case of exocytosis, which are regulated by the Rab GTPase family. miRNAs transcribed from nuclei are incorporated into the ILVs via interactions with RNA-binding proteins.

**Figure 3 membranes-12-00550-f003:**
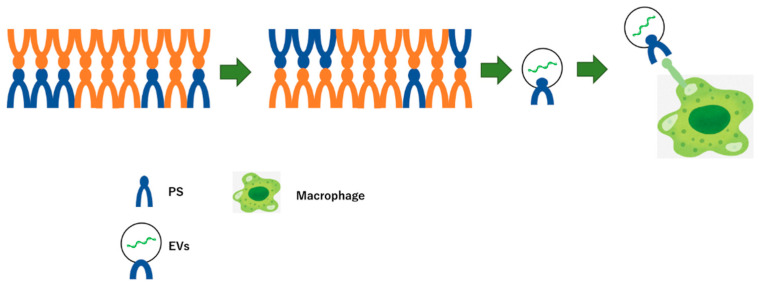
Macrophage phagocytoses apoptotic cells via externalization of phosphatidylserine (PS) on EVs induced apoptosis as “eat me signal”. In apoptotic conditions, PS in the plasma membranes of cells externalizes, which is recognize by macrophages. Further, like the plasma membrane, the surface of the EVs has the PS, leading to uptake into macrophages.

**Figure 4 membranes-12-00550-f004:**
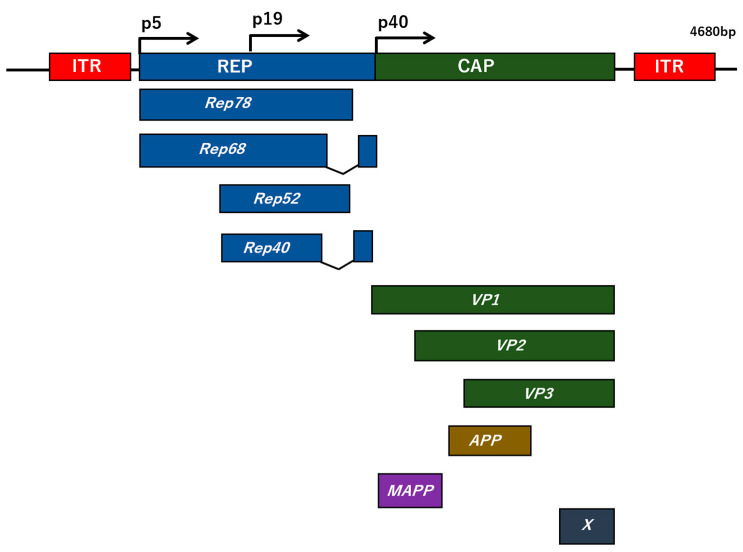
The AAV2 genome structure. The genome of AVV consists of a linear single-stranded DNA spanning approximately 4.7 kilobases (kb), flanking with two 145 nucleotides of long inverted terminal repeats (ITRs) at their terminals. Within the genome, the two viral genes *rep* and *cap* code for non-structural and structural proteins, respectively. The non-structural (*rep*) coding region is regulated by two promoters, p5 and p19, resulting in the generation of a set of four overlapping proteins, Rep78, Rep68, and Rep52, Rep40, respectively. The *cap* gene encodes three structural capsids (VP1 to VP3) by promoter p40 and the assembly-activating protein (AAP), which promotes virion assembly and the membrane-associated accessory protein (MAAP) from an alternative open reading frame (ORF). *X* ORF at the 3′ end of the *cap* gene has a specific role in AAV DNA replication.

**Table 1 membranes-12-00550-t001:** Drug-delivery-system strategies.

Pharmaceutical Technology	Advantage
Liposome	Improved transduction and safety
Microcapsule	Extension in dosing interval
PEG modification *	Extension in dosing interval
Transdermal administration	Improved convenience and safety
Sublingual administration	Immediate effect improvement
Antibody drug conjugate	Improved transduction and safety

* PEG: polyethylene glycol.

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
