# Peer review of "Extracellular Vesicles as Novel Drug-Delivery Systems through Intracellular Communications"

_membranes, 2022, doi:10.3390/membranes12060550_

Round 1

Reviewer 1 Report

The review by Y. Matsuzaka and R. Yashiro is about the employment of extra cellular vesicles (EVs) as novel drug delivery system (DDS) through intracellular communication. The authors started the review by introducing the main topics citing the most important works. The first paragraph is about the biogenesis and the release of Exosomes that I strongly suggest calling small EVs, following they described the implication of small EVs as DDS.

The text is well written, but it needs to be reorganized starting to explain what is intended with the terms “DDS through intracellular communication” and focusing all the review on this fascinating topic. I also suggest cutting the paragraph about the biogenesis because is described in hundreds of papers and reviews and focusing only on the employment of small EVs as DDS through intracellular communication factors. I suggest dividing the text in subparagraph based on the cargoes of the EVs (e.g. peptides, proteins, RNA, etc.), also a table could be useful to summarize and have a rapid overview.

Author Response

The review by Y. Matsuzaka and R. Yashiro is about the employment of extra cellular vesicles (EVs) as novel drug delivery system (DDS) through intracellular communication. The authors started the review by introducing the main topics citing the most important works. The first paragraph is about the biogenesis and the release of Exosomes that I strongly suggest calling small EVs, following they described the implication of small EVs as DDS.

Thank you very much for giving us the advice. According to reviewer’s comments, We corrected this sentence on line 46, in page 1.

The text is well written, but it needs to be reorganized starting to explain what is intended with the terms “DDS through intracellular communication” and focusing all the review on this fascinating topic.

Thank you very much for giving us the advice. According to reviewer’s comments, We corrected this sentence on line 13-15, line 21-23, line 31-35, in page 1.

I also suggest cutting the paragraph about the biogenesis because is described in hundreds of papers and reviews and focusing only on the employment of small EVs as DDS through intracellular communication factors. I suggest dividing the text in subparagraph based on the cargoes of the EVs (e.g. peptides, proteins, RNA, etc.), also a table could be useful to summarize and have a rapid overview.

Thank you very much for giving us the advice. According to reviewer’s comments, We made subparagraph in Section 2 and 3.

Reviewer 2 Report

  1. Matsuzaka et al. reviewed the role of extracellular vesicles as a novel drug delivery system through intracellular communications.  They reviewed the biogenesis and release of exosomes, EVs-mediated DDS, and EV-encapsulated adeno-associated virus and concluded. The study requires revision before it is accepted in order to make the study more impactful.

    Comments:

    1. The abstract is very vague, which requires more tangible.
    2. Why author only showed the biogenesis of exosomes (small EVs), why not microvesicles (medium or larger EVs)?
    3. "Eat me" signal and "don’t eat me" signal works in EV-based DDS? Please discuss the internalization of EVs into cells with an illustration of the mechanism.
    4. How miRNAs are sorted into EVs? Please detail with an illustration

Author Response

The abstract is very vague, which requires more tangible. Thank you very much for giving us the advice.

According to reviewer’s comments, We corrected this abstract in page.

Why author only showed the biogenesis of exosomes (small EVs), why not microvesicles (medium or larger EVs)?

Thank you very much for giving us the advice. Recently, it was reported that small EVs are useful as DDS due to their high transduction into the target cells. Therefore, we added the sentence in line 25, in page 1.

"Eat me" signal and "don’t eat me" signal works in EV-based DDS? Please discuss the internalization of EVs into cells with an illustration of the mechanism.

Thank you very much for giving us the advice. There are a possibility of “ eat me signal” on EV-based DDS. According to reviewer’s comments, we added the sentence in line 354-358, in page 9, and Figure 3 in page 9.

How miRNAs are sorted into EVs? Please detail with an illustration.

Thank you very much for giving us the advice. According to reviewer’s comments, we added the sentence on line 175-177, in page 5, and corrected figure 2.

Reviewer 3 Report

The current manuscript provides a unique but complex account of extracellular vesicles as NDDS. The topic is well chosen and the manuscript is well written. However, the articulation of the topic and the subject matter needs revision. Given that the authors pulled information from over 180 references, the authors could have provided a more detailed account of the complex terminologies used in the manuscript. A summary table of various drug delivery strategies employed by various researchers would enhance the quality of the manuscript, will cover all the relevant references, and will also provide a brief overview. The cell-cell communication part has overpowered the application aspect.

Author Response

The current manuscript provides a unique but complex account of extracellular vesicles as NDDS. The topic is well chosen and the manuscript is well written. However, the articulation of the topic and the subject matter needs revision. Given that the authors pulled information from over 180 references, the authors could have provided a more detailed account of the complex terminologies used in the manuscript. A summary table of various drug delivery strategies employed by various researchers would enhance the quality of the manuscript, will cover all the relevant references, and will also provide a brief overview. The cell-cell communication part has overpowered the application aspect. Thank you very much for giving us the advice.

According to reviewer’s comments, we added the Table 1 in page 2.

Round 2

Reviewer 1 Report

Authors answered correctly and exhaustively point by point to all the comments. The review is now fine to be accepted and publish.